# Reliability and Quality of YouTube Videos on Ultrasound-Guided Brachial Plexus Block: A Programmatical Review

**DOI:** 10.3390/healthcare9081083

**Published:** 2021-08-23

**Authors:** Noo Ree Cho, Jeong Ho Cha, Jeong Jun Park, Yun Hak Kim, Dai Sik Ko

**Affiliations:** 1Department of Anesthesiology and Pain Medicine, Gachon University Gil Medical Center, Incheon 21565, Korea; mintbit0614@gilhospital.com (N.R.C.); jeongho.car@gmail.com (J.H.C.); 2Department of Anesthesiology and Pain Medicine, CHA Bundang Medical Center, CHA University School of Medicine, Seongnam 13496, Korea; jeongjun.park@cha.ac.kr; 3Department of Biomedical Informatics, School of Medicine, Pusan National University, Yangsan 50612, Korea; yunhak10510@pusan.ac.kr; 4Division of Vascular Surgery, Department of Surgery, Gachon University Gil Medical Center, Incheon 21565, Korea

**Keywords:** ultrasonography, brachial plexus block, ultrasound-guided regional anesthesia, YouTube, learning

## Abstract

Background: Ultrasound-guided regional anesthesia has gained popularity over the last decade. This study aimed to assess whether YouTube videos sufficiently serve as an adjunctive tool for learning how to perform an ultrasound-guided brachial plexus block (BPB). Methods: All YouTube videos were classified, based on their sources, as either academic, manufacturer, educational, or individual videos. The metrics, accuracy, utility, reliability (using the Journal of American Medical Association Score benchmark criteria (JAMAS)), and educational quality (using the Global Quality Score (GQS) and Brachial Plexus Block Specific Quality Score (BSQS)) were validated. Results: Here, 175 videos were included. Academic (1.19 ± 0.62, mean ± standard deviation), manufacturer (1.17 ± 0.71), and educational videos (1.15 ± 0.76) had better JAMAS accuracy and reliability than individual videos (0.26 ± 0.67) (*p* < 0.001). Manufacturer (11.22 ± 1.63) and educational videos (10.33 ± 3.34) had a higher BSQS than individual videos (7.32 ± 4.20) (*p* < 0.001). All sources weakly addressed the equipment preparation and post-procedure questions after BSQS analysis. Conclusions: The reliability and quality of ultrasound-guided BPB videos differ depending on their source. As YouTube is a useful educational platform for learners and teachers, global societies of regional anesthesiologists should set a standard for videos.

## 1. Introduction

Regional anesthesia provides benefits for patients by reducing acute, chronic postoperative pain, postoperative nausea and vomiting, and pulmonary complications [1,2,3,4]. Ultrasound-guided regional anesthesia (UGRA) has grown in popularity over the last decade, and several advancements have led to an increase in its effectiveness and safety [5,6,7,8]. As ultrasonography (US) provides direct visualization of the needle pathway, target nerve, surrounding tissues, and local anesthetic spread around the nerve, and anesthesiologists can rapidly and more accurately perform the nerve block [9,10]. Although there is no definitive evidence that UGRA reduces peripheral nerve injury compared with the traditional nerve stimulation techniques, it has been reported that UGRA reduced the incidence of local anesthetic systemic toxicity [11,12,13] and the frequency of pneumothorax associated with US-guided supraclavicular blocks [14,15,16]. The majority of the teaching physicians believe that US increases the efficacy and safety of regional anesthesia, and recommend that UGRA be included in the teaching programs of residents and fellows [17].

The use of ultrasound is highly dependent on the operator. Despite advances in ultrasound technology, some students may find it harder to master [18,19]. Besides comprehension of the general principles of US and anatomy, UGRA requires new skills, including image interpretation, needle beam alignment, and needle trajectory-tracking [20,21]. Sites et al. [21] analyzed 520 nerve block procedures performed by anesthesia residents and found that the most common errors, such as failure to visualize the needle, need to be addressed in training programs. For procedural safety, it has been consistently suggested that other educational tools are required in addition to the conventional ultrasound workshop [22].

YouTube was established as a social platform in 2005, and has since become a popular educational platform. Rapp et al. [23] showed that YouTube was the most preferred source of surgical videos for medical students, surgical residents, and specialists. A recent survey conducted on surgeons regarding laparoscopic surgery showed that more than 86% of the trainees routinely watched online surgical videos on YouTube to learn or perfect their surgical technique [24]. Similar to laparoscopic surgery, the hand−eye−screen coordination required during UGRA requires practice, as hand and needle movements occur in three different axes, whereas the ultrasound image is only presented in two dimensions.

Several researchers have evaluated the quality of YouTube videos for medical information and skills [25,26,27,28,29,30,31,32]. They found some videos on YouTube to be educationally useful; however, a large portion of videos had a lower quality of education and some even inaccurate information [25,29,33]. This is mostly due to the lack of a review process, similar to the scientific literature publication, uncertainty of sources, and their reliability. Despite the shortcomings of YouTube videos, there is no doubt that many medical students and residents currently still seek educational information on YouTube, and that well-designed educational videos may enhance their learning, thereby serving as a highly effective educational tool [34,35,36,37]. It has been demonstrated that adding online learning methods, such as video materials, to the text learning-based pedagogical approach can improve education [37]. Thus, we aimed to investigate whether YouTube can serve as an adjunctive tool for learning the complex process of UGRA. Several studies have investigated the quality of UGRA YouTube videos. Tewfik et al. [38] showed that user-uploaded videos on YouTube had less educational characteristics than those by the anesthesia society websites. Selvi et al. [39] scored the videos of brachial plexus block (BPB) on YouTube using their own questionnaires. However, to the best of our knowledge, there has been no in-depth analysis of the quality and sources. Therefore, the objectives of this study were as follows: (1) identify the upload sources and characteristics of the YouTube videos, (2) investigate the quality of videos using three different score instruments measuring reliability and educational values, (3) examine the difference in the quality of videos between different sources, and (4) present future directions for high-quality educational videos of UGRA. We focused on the ultrasound-guided BPB, which is the most representative and common type of UGRA.

## 2. Materials and Methods

### 2.1. Search Strategy and Inclusion Criteria

YouTube (www.youtube.com) was systematically searched in July 2020 via the YouTube Data API and Google Apps Script. The code for Google App Script can be found at “https://github.com/igreg1221/YouTube-API-Search-”. We identified YouTube videos uploaded from 1 January 2005 to 31 December 2019 with the terms “brachial plexus block”, “interscalene block”, “supraclavicular block”, “infraclavicular block”, and “axillary block” in the title. Videos that demonstrated the procedures using US were included. The exclusion criteria were videos without audio or text, videos with non-English narration or captions, PowerPoint presentation slides, animations, videos that did not use the US in the procedures, procedures not performed on humans, videos only demonstrating surgical procedures, and videos with a resolution lower than 360p (480 × 360 pixels).

### 2.2. Video Metrics

Using the unique video identifier acquired from the Google Apps Script, the following video parameters were extracted using the R package “tuber” (https://CRAN.R-project.org/package=tuber (accessed on July, 2020)): (1) title, (2) video duration, (3) date of publication, (4) number of views, (5) number of likes, (6) number of dislikes, (7) number of comments, (8) resolution, (9) like ratio (Like × 100/(Like + Dislike)), (10) view ratio (number of views/days), and (11) Video Power Index (VPI) (like ratio × view ratio/100). The VPI, which was first described by Erdem et al. [40], was used to evaluate the popularity of the videos. All videos were classified based on their source, which was (1) individual (independent medical doctor without any affiliation mentioned, (2) academic (hospital or university affiliation pertaining to authors), (3) manufacturer (US manufacturer affiliation pertaining to authors), and (4) educational (corporation videos for educational purpose, i.e., The New York School of Regional Anesthesia (NYSORA), Ultrasound-Guided Regional Anesthesia and Pain Medicine (USRA)).

### 2.3. Video Accuracy, Utility and Reliability

The accuracy, utility, and reliability of each video were evaluated according to the Journal of American Medical Association Score (JAMAS) benchmark criteria on a scale of 0 to 4, as suggested by Silberg et al. [41]. The JAMAS benchmark criteria (Appendix A) consists of four individual criteria, with one point assigned for each criterion as a non-specific assessment of the source’s reliability. A score of four indicates a higher accuracy, utility, and reliability, whereas a score of 0 indicates poor accuracy, utility, and reliability.

### 2.4. Educational Value of Video

We used two scoring systems to assess the educational value of the videos. The Global Quality Score (GQS) [40,42] provides a non-specific assessment of the educational value, with five criteria (Appendix A). The Global Quality Score has a scale of 0 to 5, and a higher score indicates higher educational quality. For a more specific quality assessment of the videos, two anesthesiologists (N.R.C. and J.J.P.) who routinely perform BPB in daily practice, utilized a Brachial Plexus Block Specific Quality Score (BSQS) (Table 1) based on the Miller’s Anesthesia textbook [43], recommendations from the American and European society of regional anesthesia and pain therapy joint committee [20], and published articles [44,45,46,47]. To assess the extent of the procedure-specific knowledge and skills the videos contained, we evaluated the BSQS from the procedure preparation to post-procedural steps. The BSQS consisted of pre-procedure, equipment preparation, intra-procedure, and post-procedure scores. All details are provided in Table 1. Briefly, for pre-procedure scores, videos mentioned the approaches and indicated the procedures, targeted dermatome, and the patient’s position. For equipment preparation, the equipment for ultrasound-guided BPB with aseptic fashion was assessed. For the intra-procedure scores, videos showed the sonographic view of the procedure with anatomical landmarks, needling techniques, and needle tip confirmation. Lastly, for post-procedure scores, the videos mentioned how to verify the success of procedures and types of complications. Of the 16 criteria, five indispensable steps were assessed, and the two scores were weighted. The videos were given 1 point if the instructions were presented orally or in text. All scorings were performed independently by two authors. Videos with different scores were reassessed until a consensus was reached. To visualize the BSQS scores according to video sources, we have shown the percentage of BSQS questions as a heatmap.

### 2.5. Statistical Analysis

The R program (Version 3.6.0, R Foundation for Statistical Computing, Vienna, Austria) was used for the statistical analyses. Descriptive statistics were used to quantify the video characteristics. As the parameters did not show a normal distribution, the Kruskal−Wallis test was used for the intergroup comparisons and the Mann−Whitney U test was used to identify the group that caused the difference. All data are presented as mean ± standard deviation (median). The significance level was set at *p* < 0.05. Spearman rank correlation was used to identify the correlations between the variables. Cohen ƙ and intraclass correlation coefficients were calculated to evaluate the degree among the raters using R package “irr” (https://cran.r-project.org/web/packages/irr/index.html (accessed on August 2020)) with a single-measurement, absolute-agreement, and two-way mixed effects model [48].

## 3. Results

### 3.1. Overall Video Metrics

The five search terms yielded 799 unique video identifiers through the YouTube Data API and Google Apps Script (Figure 1). After excluding 170 duplicates, 629 videos were watched in full detail. A total of 444 videos met the exclusion criteria, and the most common reason for exclusion was the lack of audio or captions in the videos (*n* = 217). Finally, 175 videos were analyzed for their quality assessment. Overall, 50.3% (*n* = 88) of the videos were individual, 20.6% (*n* = 36) were academic, 10.3% (*n* = 18) were manufacturer, and 18.9% (*n* = 33) were educational videos (Table 2). All variables of 175 videos, including the unique identifier, BSQS, GQS, JAMAS, and VPI, among others, are provided in the Appendix A. The mean video duration was 247.67 ± 208.40 (191) s. The mean number of views was 18,907.94 ± 46,373.86 (785) times. The mean view ratio was 7.73 ± 16.78 (0.53). The mean number of days since upload was 1948.99 ± 1079.98 (1883) days. The mean number of comments was 1.38 ± 3.15 (0). Videos received an average of 50.30 ± 121.30 (3) likes and 3.21 ± 7.20 (0) dislikes, with a mean-like ratio of 92.33 ± 17.28 (100). The mean VPI was 8.82 ± 17.38 (0.88), JAMAS was 0.71 ± 0.82 (1), and the GQS and BSQS scores were 1.68 ± 0.74 (2) and 8.58 ± 4.03 (9), respectively. The JAMAS, GQS, and BSQS scores were determined through consensus when the scores from two anesthesiologists showed any discrepancy. Before consensus, the intraclass correlation coefficients calculated for BSQS, GQS, and JAMAS were 0.862 (95% confidence interval (CI)-0.818 to 0.895), 0.802 (95% CI-0.742 to 0.849), and 0.883 (95% CI-0.844 to 0.912).

The numbers of views, likes, dislikes, comments, and VPI were significantly different across the sources (Table 2). The count of views was highest for manufacturer videos and statistically higher than that of individual and academic videos (*p* < 0.001). Educational videos had more likes than individual and academic videos (*p* < 0.001). Manufacturer and educational videos had more dislikes than individual videos (*p* < 0.001). The VPI, as a measurement of popularity, was higher on the manufacturer and educational videos than that for the individual videos (*p* < 0.001). The length of the video was not different among sources (*p* = 0.32). The JAMAS benchmark criteria were used to assess the accuracy, utility, and reliability. Academic (1.19 ± 0.62), manufacturer (1.17 ± 0.71), and educational videos (1.15 ± 0.76) had higher JAMAS scores than individual videos (0.26 ± 0.67) (*p* < 0.001). The GQS and BSQS were used for the non-specific and specific assessment of educational quality, respectively. Manufacturer (11.22 ± 1.63) and educational videos (10.33 ± 3.34) had a higher BSQS than individual videos (7.32 ± 4.20) (*p* < 0.001). The mean GQS did not statistically differ based on the video source (*p* = 0.18). In the correlation analysis of BSQS, GQS, and JAMAS, BSQS and GQS showed a strong correlation (ρ = 0.74, *p* < 0.001) (Multimedia Appendix A). Scores and other quantitative variables (view, like, dislike, VPI, among others) showed a positive correlation; however, the like ratio (Like × 100 / (Like + Dislike)) showed a negative correlation with all other variables.

### 3.2. Further Analysis of Reliability and Quality Assessment Scores

We determined the percentage of BSQS questions that were addressed per source and depicted it as a heatmap (Figure 2). Most notably, all sources weakly addressed the equipment preparation (Q4–Q8) and post-procedure questions (Q15–16). The intra-procedure questions (Q9–Q14) were the most addressed. To investigate the difference by year for BSQS, GQS, and JAMAS, we analyzed the changes in the scores by year according to the sources (Figure 3). For BSQS and GQS, no apparent changes were observed for all sources. In JAMAS, all videos except individual videos showed a tendency for incline by year.

The decline of BSQS of individual and academic videos by year was significantly lower than that of the manufacturer and educational videos. Due to an increase in awareness of intellectual property rights and the reinforcement of restrictions, the standardized reporting system has become more common when uploading medical information on YouTube. As a result, the JAMAS score seems to have increased by year on all videos except the individual videos.

## 4. Discussion

We analyzed the educational quality of BPB videos on YouTube and found that manufacturer and educational videos had higher scores in all aspects; however, all sources had common deficiencies in their contents and did not increase the learner’s participation. The coronavirus disease-19 (COVID-19) pandemic has led to a reduction in the number of elective surgeries being performed. As a result, the opportunity for fellows to practice regional anesthesia has also been reduced [49,50,51]. Moreover, due to social distancing and other local policies regarding the size of meetings or gatherings, conferences and hands-on courses have been canceled or substituted for remote video and audio conferences. Although there is no substitute for expert guidance by experienced instructors in clinical practice, a new learning tool may be required to compensate for the lack of conventional education in this era [52]. Many universities have launched their channels, including medicine and science, encouraging students and teachers to cultivate a student−teacher coordinated effort and enable real-time feedback from students [53,54]. To this end, we investigated the educational quality of YouTube videos demonstrating a BPB by performing a systematic search and applying multiple scoring systems.

When assessing the source, the manufacturer and educational videos had a higher BSQS, GQS, JAMAS score, and VPI compared with individual videos. In contrast, academic videos only had a higher JAMAS score compared with individual videos. The reliability and quality of the manufacturer videos were better than that of the individual ones; they focused on US probe usage and intra-procedure techniques, such as needle tip visualization. Despite these limitations, for most manufacturer’s videos, well-informed experts performed and explained the procedure. The educational videos were mostly uploaded by NYSORA and Ultrasound for Regional Anesthesia (URSA). These websites are well-organized with highly accurate educational content; novice trainees can learn basic anatomy and clinical skills for UGRA from NYSORA and URSA [55,56,57,58,59,60,61]. We observed several common features in all videos; (1) the BSQS lacked the equipment preparation and post-procedure information and (2) there were very few comments (1.38 ± 3.15 (median: 0)). We weighted the two scores on all questions in post-procedure scores as observing success and complications is as important as the procedure itself. However, most videos, regardless of their source, scarcely mentioned the post-procedural information. These common deficiencies should be addressed when making ultrasound-guided BPB videos. Furthermore, the lack of comments indicates that videos did not attract students’ participation and may not play an active role on social media outreach.

The influence of videos in medical education is likely to increase. Studies have pointed out the benefits of multimedia-enhanced teaching; it significantly improves surgical performance and understanding of complex temporal and spatial events [62,63,64]. Furthermore, various trials based on video-based coaching have been performed to improve the surgical techniques [65,66,67,68]. Mota et al. [69] investigated the difference in the characteristics of video usage between residents and specialists. Interestingly, they showed that residents used YouTube more significantly than the specialists. They preferred a more easily accessible information tool, with feedback, comments, and various points of view on each topic; several of these are the main advantages of YouTube [70]. With increasing access to the Internet and the widespread use of mobile devices, trainees with limited access to information on novel techniques and technologies may also be able to access the instructional material. However, these videos do not undergo a peer-review process and are only screened for copyright infringement, not for their educational value and quality. Many authors reported poor video quality, inaccurate information, incomprehensible or lack of audio, and the lack of background patient information on videos for common surgical procedures such as appendectomy and cholecystectomy [71]. Many authors agree to use YouTube videos as an educational tool; however, prior to utilizing these videos, they caution against misleading information [32]. Urgent guideline advocacy is required for publishing educational videos from each society of medical specialties [72].

Despite these drawbacks of YouTube videos, the affordability, easy access, and the ability to interact with the global community firmly establishes YouTube as an educational learning tool. In the COVID-19 pandemic era, the challenging conditions will undoubtedly minimize the resident teaching and exposure to regional anesthesia [73,74]. A dramatic drop in exposure to regional anesthesia training among anesthesia residents may potentially lead to negative effects in the future. Medina et al. [75] suggested strategies such as watching didactic material and high-quality videos to maintain relevant education and training in regional anesthesia procedures.

Our study programmatically reviewed ultrasound-guided BPB videos on YouTube and evaluated the reliability and quality of videos. YouTube videos on UGRA procedures must be validated, and the global community of regional anesthesiologists should play an active role. Our reporting on the common deficiencies in the contents of videos can help improve the future video quality of regional anesthesia techniques. Moreover, using the social media platform, communication between uploaders and learners should be encouraged to maximize the effectiveness of learning on YouTube. YouTube has the potential to be the largest educational platform; therefore, regional anesthesiologists worldwide should assess the quality of these videos and promote effective communication through YouTube. Our results can be applied to any field of medical education, especially those that require repetitive practices and hand−eye−screen coordination, such as hybrid (open and endovascular) vascular surgery. Our study has limitations. Metrics, such as likes, dislikes, and the length of the video do not fully represent the viewer’s response. The average percent viewed may show how long viewers watched the total length of the video; however, YouTube Data API does not provide these due to privacy issues. We were unable to conclude that high-scored videos helped viewers improve their knowledge and skills. In a future study, this should be implemented to make a video that satisfies all the scores we used, and then release it on YouTube to see if it helps viewers improve UGRA learning.

## Figures and Tables

**Figure 1 healthcare-09-01083-f001:**
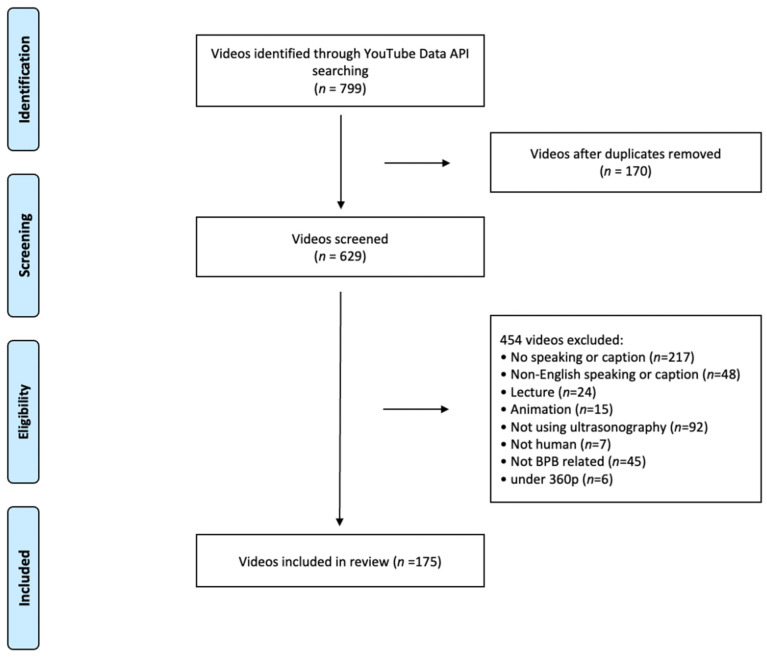
Flowchart of the video search and screening process: video metrics, reliability, overall quality, and specific quality of videos by source.

**Figure 2 healthcare-09-01083-f002:**
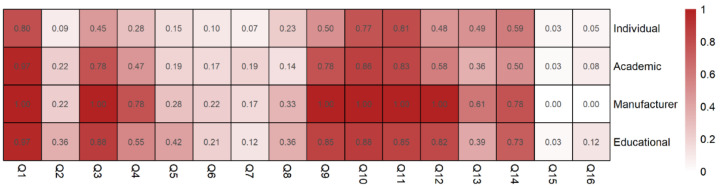
Heatmap of brachial plexus block specific quality scores according to video sources. The *X*-axis represents each question and *Y*-axis represents the video source. The number in the box represents the percentage of questions addressed. The color is intensified as the percentage increases.

**Figure 3 healthcare-09-01083-f003:**
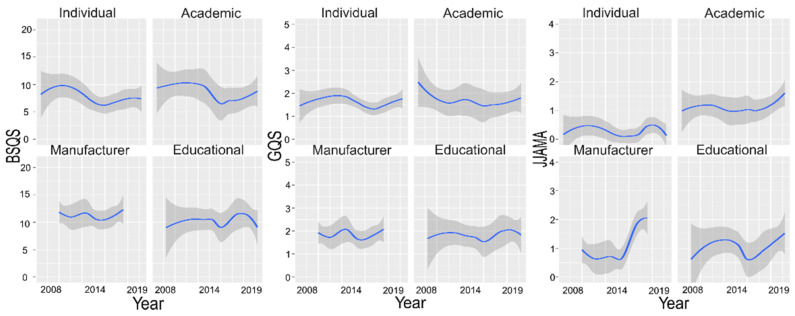
Changes of BSQS, GQS, and JAMAS by year according to sources. The smoothed loess regression represents the changes of scores from 2005 to 2019.

**Table 1 healthcare-09-01083-t001:** Brachial plexus block specific quality score.

	Brachial Plexus Block Specific Quality Score	Score
Pre-Procedure Scores		
	Q1. Mentioned which procedure	1
	Q2. Mentioned targeted dermatome or indication	2
	Q3. Patient position	1
	Equipment preparation	
	Q4. High-frequency linear probe	1
	Q5. Disinfectant solution	1
	Q6. Local anesthetics (which and how much)	1
	Q7. Needle gauze	1
	Q8. Sterile gel and sterile probe cover	1
Intra-Procedure Scores		
	Q9. Probe placement	1
	Q10. Anatomical landmarks	2
	Q11. Important vessels and structures	1
	Q12. Needling technique; in-plane or out-of-plane	1
	Q13. Needle tip confirmation (negative aspiration or small amount injection or nerve stimulation use)	2
	Q14. Spread of local anesthetics	1
Post-Procedure Scores		
	Q15. Dermatome check or nerve stimulation use	2
	Q16. Complications	2
	Total	21

**Table 2 healthcare-09-01083-t002:** Video metrics, reliability, overall quality, and specific quality of the videos based on the source.

	Individual	Academic	Manufacturer	Educational	*p*-Value	Post Hoc. Tukey’s Test
Videos, *n* (%)	88 (50.29)	36 (20.57)	18 (10.29)	33 (18.86)		
Years, median (min, max)	2015 (2008, 2019)	2015 (2008, 2019)	2014 (2010, 2017)	2015 (2009, 2019)		
Video metrics, mean ± SD (median)						
Views	3472.78 ± 10,333.14 (458)	21,120.58 ± 47,242.49 (509)	47,873.61 ± 87,483.11 (9675)	41,855.12 ± 56,751.61 (19,418)	<0.001	Manufacturer > Individual, Academic; Education > Academic
Likes	10.95 ± 33.80 (2)	31.11 ± 69.70 (2)	119 ± 205.87 (18)	138.70 ± 183.52 (45)	<0.001	Educational > Individual, Academic
Dislikes	0.74 ± 2.35 (0)	3.44 ± 6.81 (0)	7.5 ± 13.03 (2)	7.21 ± 9.05 (4)	<0.001	Manufacturer, Educational > Individual
Comments	0.59 ± 1.64 (0)	1.44 ± 2.67 (0)	2.11 ± 4.78 (0)	3.03 ± 4.69 (1)	<0.001	Educational > Individual
Length (seconds)	247.65 ± 212.64 (193)	238.56 ± 237.33 (166)	195.89 ± 72.47 (162)	285.94 ± 213.82 (234)	0.32	
VPI	1.59 ± 3.35 (0.38)	7.80 ± 14.32 (0.82)	20.81 ± 25.28 (13.19)	21.35 ± 24.75 (10.49)	<0.001	Manufacturer, Educational > Individual
Reliability, mean ± SD (median)						
JAMAS	0.26 ± 0.67 (0)	1.19 ± 0.62 (1)	1.17 ± 0.71 (1)	1.15 ± 0.76 (1)	<0.001	Academic, Manufacturer, Educational > Individual
Overall quality, mean ± SD (median)						
GQS	1.58 ± 0.71 (1)	1.69 ± 0.89 (1)	1.83 ± 0.38 (2)	1.85 ± 0.82 (2)	0.1087	
Specific quality, mean ± SD (median)						
BSQS	7.32 ± 4.20 (7)	8.72 ± 3.84 (8.5)	11.22 ± 1.63 (11)	10.33 ± 3.34 (10)	<0.001	Manufacturer, Educational > Individual

BSQS—Brachial Plexus Block Specific Quality Score; GQS—Global Quality Score; JAMAS—Journal of American Medical Association; SD—standard deviation; VPI—Video Power Index.

## Data Availability

The data that support the findings of this study are available from the corresponding author upon reasonable request.

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
