# Peer review of "Reliability and Quality of YouTube Videos on Ultrasound-Guided Brachial Plexus Block: A Programmatical Review"

_healthcare, 2021, doi:10.3390/healthcare9081083_

Round 1

Reviewer 1 Report

The manuscript presents a study about the reliability and quality of a sample of YouTube videos on ultrasound-guided brachial plexus block. The study provides an important perspective on how to assess quality of videos that were not necessarily produced for education, but which can be used to complement formal learning. I have a number of concerns that I would like to share with the authors that may be used for revision.

  • I suggest to expand the theory about learning and, specifically, to address differences and complement between formal learning and informal learning. I believe that it is important to highlight learning benefits of YouTube dealing with the diverse learning paradigms.
  • I recommend to check punctuation in the Abstract “The metrics, accuracy, utility, and reliability (using the Journal of 24 American Medical Association Score benchmark criteria (JAMAS)), educational quality (using the 25 Global Quality Score (GQS), and Brachial Plexus Block (BPB) Specific Quality Score (BSQS)) were 26 validated”.
  • In lines 88-90 (page 2) reference is made to three different score instruments, but the reader does not yet know which they are because they are not described until later.
  • While it is understandable that the search started in 2005, it is less clear why it was restricted to 31 December 2019 since YouTube was systematically searched in July 2020.
  • I would like to see more information about how inter-rater reliability was calculated for the diverse coding processes (which measures resulted in using Cohen’s k).
  • There is a general problem of alignment with the figures. In addition, the caption of figure 2 is truncated.
  • I suggest providing information on the temporal distribution of the analysed videos.
  • I think there is a problem with the flowchart in figure 1. “Videos after duplicates removed” is lower than the number of videos analysed.
  • Finally, I suggest expanding the Discussion and adding the limitations of the study. Further directions for future research should be in the Conclusions, a section I suggest adding.

Author Response

Reviewer 1

Comments and Suggestions for Authors

The manuscript presents a study about the reliability and quality of a sample of YouTube videos on ultrasound-guided brachial plexus block. The study provides an important perspective on how to assess quality of videos that were not necessarily produced for education, but which can be used to complement formal learning. I have a number of concerns that I would like to share with the authors that may be used for revision.

  1. I suggest to expand the theory about learning and, specifically, to address differences and complement between formal learning and informal learning. I believe that it is important to highlight learning benefits of YouTube dealing with the diverse learning paradigms.

- Thank you for your kind suggestion. Adding online education, such as video materials, to classic education based on textbooks has been widely studied, and many studies have shown their superiority. We have expanded our theory according to your comment. Thank you for strengthening the rationale of our study

Line 81: It has been demonstrated that adding online learning methods, such as video materials, to the text learning-based pedagogical approach can improve education.

  1. I recommend to check punctuation in the Abstract “The metrics, accuracy, utility, and reliability (using the Journal of 24 American Medical Association Score benchmark criteria (JAMAS)), educational quality (using the 25 Global Quality Score (GQS), and Brachial Plexus Block (BPB) Specific Quality Score (BSQS)) were 26 validated”.

- Thank you for your indication. We have corrected it accordingly.

Line 24: The metrics, accuracy, utility, reliability (using the Journal of American Medical Association Score benchmark criteria [JAMAS]), and educational quality (using the Global Quality Score [GQS] and Brachial Plexus Block Specific Quality Score [BSQS]) were validated.

  1. In lines 88-90 (page 2) reference is made to three different score instruments, but the reader does not yet know which they are because they are not described until later.

- We mean that the three different scores are JAMAS, GQS, and BSQS. We have corrected it to improve readability accordingly.

Line 90: (2) investigate the educational quality of videos using three different score instruments measuring reliability and educational values;

  1. While it is understandable that the search started in 2005, it is less clear why it was restricted to 31 December 2019 since YouTube was systematically searched in July 2020.

- We understand your concern. We suspect that videos should be available publicly for at least 6 months, and metrics such as views, comments, and VPI be analyzed. Please consider our inclusion criteria.

  1. I would like to see more information about how inter-rater reliability was calculated for the diverse coding processes (which measures resulted in using Cohen’s k).

- Thank you for your suggestion. We have added the details about the package and model.

Line 166: Cohen Æ™ and intraclass correlation coefficients were calculated to evaluate the degree among the raters using R package ‘irr’ (https://cran.r-project.org/web/packages/irr/index.html) with a single-measurement, absolute-agreement, and 2-way mixed effects model.

  1. There is a general problem of alignment with the figures. In addition, the caption of figure 2 is truncated.

- Thank you for your suggestion. We suspect that by adding figures in the manuscript, the alignment of figures looks inappropriate. However, in the printed or epub version, they will be aligned properly. Accordingly, we have corrected the truncated caption.

  1. I suggest providing information on the temporal distribution of the analysed videos.

I think there is a problem with the flowchart in figure 1. “Videos after duplicates removed” is lower than the number of videos analysed.

- Thank you for your kind suggestion. We have added information in Table 2 about the median year (min-max) of the videos according to their sources. Kindly review it.

- We have found the flaw in figure 1, as you mentioned. We have corrected it accordingly. Thank you for your suggested correction.

  1. Finally, I suggest expanding the Discussion and adding the limitations of the study. Further directions for future research should be in the Conclusions, a section I suggest adding.

- Thank you for your kind suggestion. We have added the limitations and future research direction. As a result of your comments, we could improve our manuscript substantially.

Line 327: Our study has limitations. Metrics, such as likes, dislikes, and the length of the video do not fully represent the viewer’s response. The average percent viewed may show how long viewers watched the total length of the video; however, YouTube Data API does not provide these due to privacy issues. We were unable to conclude that high-scored videos helped viewers improve their knowledge and skills. In a future study, this should be implemented to make a video that satisfies all the scores we used, and then release it on YouTube to see if it helps viewers improve UGRA learning.

Reviewer 2 Report

Dear authors,

I read the manuscript with great interest.

Due to COVID-19, the education has been been restricted all over the world. As a result, video education was particularly active in 2019, and I feel that videos are clearly more effective than textbooks for understanding medical procedure.

Many students must be watching YouTube, so I think it makes sense to focus on YouTube.

I think the explanation in Fig.3 is insufficient. What indicate the horizontal and vertical axes represent? Score and the year?

I couldn’t understand the difference between academic and educational videos. Please explain the difference between the two in a little more detail.

I felt that it didn't make much sense to compare academic / educational videos with individual videos. Do you suggest that individual videos are of poor quality and should not be watched?

As for the quality of the video, the researchers watched and scored (using criteria) the video?

What should readers read from Figure 3?

Table 2

As an index to measure the quality of a video, I don't think it is appropriate using the length of the video. Rather, I think that the viewer retention rate is necessary.

This research focuses on local anesthesia. Can this research be applied to other fields? Abstract explanations are given in the discussion, but I think it would be better if there were explanations using concrete examples.

Author Response

Reviewer 2

Comments and Suggestions for Authors

I read the manuscript with great interest.

Due to COVID-19, the education has been been restricted all over the world. As a result, video education was particularly active in 2019, and I feel that videos are clearly more effective than textbooks for understanding medical procedure.

Many students must be watching YouTube, so I think it makes sense to focus on YouTube.

  1. I think the explanation in Fig.3 is insufficient. What indicate the horizontal and vertical axes represent? Score and the year?

- We agree that there is insufficient information in Figure 3. X-axis means year and Y-axis means scores of each scoring system (BSQS, GQS, and JAMA). We have modified it for readability. Kindly let us know if it is not enough.

  1. I couldn’t understand the difference between academic and educational videos. Please explain the difference between the two in a little more detail.

- We defined academic videos as videos that authors affiliated with hospitals or universities made and educational videos as corporation videos for educational videos, like NYSORA. As our results revealed that educational videos scored better than others in the scoring system, NYSORA provides high-quality educational videos of UGRA. Among NYSORA videos, most presenters are doctors who work in the hospital affiliated with the university. However, only a few videos are uploaded of NYSORA videos on YouTube, and most of them are only accessible when paid on their website. We explored a question regarding whether non-profit university-affiliated authors made videos that were as high quality as those made by profit-based corporation. Kindly let us know if this explanation is insufficient.

  1. I felt that it didn't make much sense to compare academic / educational videos with individual videos. Do you suggest that individual videos are of poor quality and should not be watched?

- As you pointed out, the individual videos scored lower than other videos. Even though individual videos had relatively more deficiencies in the content, common deficiencies were found in videos of all groups. We suggest from our result that when making videos of UGRA, it will be helpful to consider the common deficiencies so that videos give viewers more help to improve learning.

  1. As for the quality of the video, the researchers watched and scored (using criteria) the video?

- Yes, two anesthesiologists (Noo Ree Cho and Jeong Jun Park), who are co-authors of this manuscript, watched all the included videos and scored them independently. Videos with different scores were reassessed until a consensus was reached. We described it in the Method (subheading: Educational Value of Video) on the first submission. Kindly let me know if additional information is necessary.

  1. What should readers read from Figure 3?

- We investigated the difference by year for BSQS, GQS, and JAMAS according to the sources of whether video quality improves over the years. Of the scores, only JAMAS seems to have increased by year on all videos except individual videos; however, for BSQS and GQS, no apparent changes were observed for all sources. It may present that the quality of videos does not improve over the years, and the global community of regional anesthesia needs to play an active role in making high-quality educational videos. We described it in the Method (subheading: Further analysis of Reliability and Quality Assessment Scores) and Discussion sections on the first submission. Kindly let me know if additional information is necessary.

  1. Table 2, As an index to measure the quality of a video, I don't think it is appropriate using the length of the video. Rather, I think that the viewer retention rate is necessary.

- Thank you for your suggestion. We agree that the viewer retention rate truly reflects viewers’ preferences. Unfortunately, to our best knowledge, viewer retention rates (also known as average percent viewed) are only provided to videos’ owners. Analytics on YouTube Studio (http://studio.youtube.com) are only accessible to the channel’s owner. We have added it to the Discussion section as a limitation.

Line 327: Our study has limitations. Metrics, such as likes, dislikes, and the length of the video do not fully represent the viewer’s response. The average percent viewed may show how long viewers watched the total length of the video; however, YouTube Data API does not provide these due to privacy issues. We were unable to conclude that high-scored videos helped viewers improve their knowledge and skills. In a future study, this should be implemented to make a video that satisfies all the scores we used, and then release it on YouTube to see if it helps viewers improve UGRA learning.

  1. This research focuses on local anesthesia. Can this research be applied to other fields? Abstract explanations are given in the discussion, but I think it would be better if there were explanations using concrete examples.

- Thank you for your suggestion. We agree that examples should be mentioned in the conclusion. We added it to the Discussion.

Line 325: Our results can be applied to any field of medical education, especially those that require repetitive practices and hand-eye-screen coordination, such as hybrid (open and endovascular) vascular surgery.